# Development and piloting of a highly tailored digital intervention to support adherence to antihypertensive medications as an adjunct to primary care consultations

Aikaterini Kassavou, Vikki Houghton, Simon Edwards, James Brimicombe, Stephen Sutton

Department of Public Health and Primary Care, School of Clinical Medicine, University of Cambridge, Cambridge, UK

**Correspondence to**
Dr Aikaterini Kassavou;
kk532@medschl.cam.ac.uk

## ABSTRACT

**Objectives** This paper describes the systematic development and piloting of a highly tailored text and voice message intervention to increase adherence to medication in primary care.

**Methods** Following the Medical Research Council guidance, this paper describes (a) the systematic development of the theoretical framework, based on review of theories and meta-analyses of effectiveness; (b) the systematic development of the delivery mode, intervention content and implementation procedures, based on consultations, face-to-face interviews, think-aloud protocols, focus groups, systematic reviews, patient and public involvement/engagement input, intervention pre-test; and (c) the piloting of the intervention, based on a 1-month intervention; and follow-up assessment including interviews and questionnaires. The mixed-methods analysis combined findings from the parallel studies complementarily.

**Results** intervention development suggested the target behaviour of the intervention should be the tablets taken at a regular time of the day. It recommended that patients could be more receptive to intervention content when they initiate medication taking or they change prescription plan; and more emphasis is needed to patients' consent process. Intervention piloting suggested high intervention engagement with, and fidelity of, the intervention content; which included a combination of behaviour change techniques, and was highly tailored to patients' beliefs and prescription plan. Patients reported that the intervention content increased awareness about the necessity to take and maintain adherent to medication, reinforced social support and habit formation, and reminded them to take medication as prescribed.

**Conclusion** Tailored automated text and voice message interventions are feasible ways to improve medication adherence as an adjunct to primary care.

**Trial registration number** ISRCTN10668149.

### Strengths and limitations of this study

► This is the first medication adherence intervention for patients with hypertension and comorbidities that has been developed and piloted within the UK primary care setting.
► The study used rigorous methodology to collect and analyse data from multiple perspectives.
► This study involved and engaged patients and public at all stages of the intervention development and piloting.
► Future studies should include real-time and objective measures of the intervention determinants and outcomes.

in England,[1] and often accompanies other conditions, including type 2 diabetes mellitus (T2DM), coronary heart disease and stroke.[2] These conditions are major risk factors for disability and premature death,[2] and medication adherence can significantly lower these risks.[1] For example, in England over 10 years, an estimated 7000 quality of life-years (QALYs) could be saved and £120 million not spent on related health and social care, if people had better adherence to antihypertensive medications.[3]

Adherence is defined as taking at least 80% of the prescribed tablets and/or having medications dispensed for at least 95% days of a prescribed period.[4] However, substantial proportions of patients do not take their medication as prescribed.[5] A recent meta-analysis showed that 40% of people do not adhere adequately to cardiovascular medication and the percentage was similar (41%) for non-adherence to antihypertensive medications.[6] Taking into consideration the growing prevalence of HBP, comorbidities and the ageing population,[7] especially in

## INTRODUCTION

Hypertension or high blood pressure (HBP) affects approximately one in four people

lower and middle-income countries,[8] it is likely that there will be an increase in demand for healthcare resources to support medication adherence. The National Institute for Health and Care Excellence (NICE) recommends the development of novel and cost-effective interventions to aid patients' decision-making about taking medicines as an adjunct to healthcare providers' consultations.[9 10]

Tailored mHealth interventions, such as text (SMS) and voice (IVR) messaging intervention, is one way to support patients' adherence between their consultations. Review evidence found that such interventions can effectively support adherence to different types of medication, including antihypertensive tablets, compared with usual care only,[11 12] and can potentially be cost-effective if applied to a large number of people.[13] For example, a recent study suggested that such interventions could result in a 2.3% increase in QALYs and overall savings of $A21 120 during a patient's lifetime.[14] However, no such intervention has been developed and piloted in the UK primary care setting.

The aim of this paper is to describe the development and piloting of a highly tailored text and voice messaging intervention to support adherence to antihypertensive tablets. Although several guidelines have been proposed to the development of mHealth interventions,[15–17] this paper has been informed by the Medical Research Council (MRC) Framework[18] as it leads not only the process to develop the intervention content and delivery mode, but also the implementation procedures within the primary care National Health Service (NHS), which is an important process in developing and piloting novel interventions.

## METHODS

Review of theory and evidence, interviews, focus groups, consultations and patients and public (PPI) involvement and engagement (PPE) informed the intervention development process (see online supplementary appendix 1); and a 1-month pre–post intervention study, with follow-up assessment informed the piloting of the intervention (see online supplementary appendix 2).

### Intervention development

#### Step 1: systematic development of the tailored intervention

The systematic development of the tailored intervention was informed by the guidance provided by Dijkstra and De Vries[19] and Kreuter and colleagues[20] on the process of developing computer telephony interventions. This process included the development of the theoretical framework, tailoring and integration.

#### Develop the theoretical framework

The theoretical framework that distinguishes between intentional non-adherence (INA) and non-intentional non-adherence (NINA) guided the development of this intervention.[21] To address INA and NINA, we identified the empirical evidence of the proximal and modifiable determinants of adherence to medication[22] and mapped them into INA and NINA framework. Following that, we used the taxonomy of the behaviour change techniques (BCTs)[23] to identify the techniques that impact on INA and NINA determinants (see online supplementary appendix 3). Our decision on the BCTs for this intervention was informed by findings from two systematic literature reviews[11 12] and previous empirical evidence.[24]

#### Tailor the theoretical determinants

To develop a highly tailored intervention, we produced a tailoring matrix, which consists of a theory-based questionnaire, an algorithm of decision rules, the schedule and the message file. The theory-based questionnaire includes items measuring each participant's values of the theoretical determinants. The algorithm includes decision rules (ie, cut-off values), which facilitates the decisions on what message to prioritise and indicates a default feedback. The tailoring process of the intervention messages is informed by multiple sources, which can be grouped in three 'clusters': the first cluster of tailoring is based on participant's unique characteristics (eg, name) and is informed by records from the primary care practices; it aims to gain person's attention to the intervention by making messages personally relevant. At the second cluster of tailoring we use data from participant's responses at the tailoring questionnaire and their prescription plan. This cluster includes two sublevels of tailoring: tailoring based on the degree of (a) INA and NINA, and (b) each of the modifiable INA and NINA theoretical determinants. At the third cluster, we tailor the intervention to participants' responses and feedback during the intervention. Participants are able to further tailor the intervention content by requesting a higher or lower degree of information for a particular determinant or by requesting feedback on a different determinant. Intervention tailoring at this cluster takes into consideration participants' navigation options until the request. The schedule includes the frequency of the messages for the duration of the intervention, which participants can increase, reduce or stop. The message file included the tailored BCTs.

#### Integrate theory to intervention

The frequency, combination and sequence of the BCTs were integrated into one intervention. The decision on the above elements was based on the phase participants are in (eg, uptake or maintain), having as the starting point the collection of the medication from the dispensary.

#### Step 2: development of the intervention delivery mode, content and implementation procedures

This step of the intervention development aimed to convert the theoretical underpinnings into features of the delivery mode, intervention content and implementation procedures, and it involved the development of the delivery mode, the pre-test of the theoretical framework

and the delivery mode, the pre-test of the delivery mode and the theoretical determinant, and the development of the implementation procedures within primary care.

### Develop the intervention delivery mode

To translate the theoretical underpinning into features of the intervention delivery mode, consultations with six experts from the industry and academia were conducted. Meetings with telecommunication and IT experts in the University of Cambridge informed the decisions about the development of the IVR application. The discussions informed the development of features of the IVR applications, which involved the navigation options, the degree of interactivity, the voice delivering the messages and the prompts to facilitate participants' navigation and dialogue. The discussions for the development of the SMS application involved the necessary features to accommodate messages on complex prescription plans. A custom written database application contained the functionality to create output for both delivery modes, and data were transferred at regular intervals to both platforms (IVR and SMS) using secure data hosting protocols. The digital platform could also facilitate participants' feedback during the intervention or at a different time (ie, by triggering an inbound call). Voice recognition facilitated interaction and data collection. The digital platform enabled flexible schedule of messages delivery; for example, messages could be delivered at times tailored to participant's prescription plan (eg, 1–4 messages per day).

### Pre-test the acceptability of the theoretical framework and delivery mode

One interview study and PPI and PPE informed the acceptability of the theoretical framework and delivery mode.

Face-to-face, in-depth interviews were conducted with patients and healthcare providers to assess the acceptability of the delivery mode, the theoretical framework and generate relevant content. Patients were randomly identified and selected by the practice manager from practice databases. Primary care practices within the Cambridgeshire and Peterborough Clinical Commissioning Group (CCG) were eligible for inclusion, and five primary care practices from different areas of deprivation (based on Index of Multiple Deprivation (IMD)) took part in the study. Patients were eligible if they (a) had a diagnosis of HBP, T2DM or both health conditions; (b) had been prescribed at least one antihypertensive medication and/or glucose-lowering medication as confirmed by practice records for at least 3 months before recruitment; (c) had poorly controlled blood pressure and/or glucose levels as indicated by practice records, or had gaps in collecting repeat prescriptions; and (d) were aged 18 years or older. Patients were excluded when they (a) had hearing or speaking impairment, (b) had a diagnosis of dementia, aphasia or other cognitive difficulties that could affect the study participation, (c) had a

recent severe life-threatening event, (d) had difficulty in speaking or understanding English, or (e) participated in another study. A practice general practitioner double screened the list of selected patients against the inclusion and exclusion criteria to confirm eligibility. Eligible patients were invited by post or telephone calls following up a postage invitation. Healthcare providers (ie, nurses and healthcare assistants) who had experience with advising patients about their medications were eligible to participate and interviewed in the practices. Nineteen patients and five healthcare providers were interviewed face-to-face. Both patients and healthcare providers' interview data informed inductively the content of the theoretical determinants and the BCTs, and provided their views on the acceptability of the automated intervention within the primary care setting.

PPI and PPE was conducted to assess acceptability of the delivery mode and the theoretical framework, and inform the intervention content. PPI/E members were recruited opportunistically using emails or adverts. In total, 100 PPI/E members took part in three events: one meeting with a diabetes patients' group (n=20), one open event at Addenbrookes' Hospital (n=30) and one open event at the Cambridge Science Festival (n=50). During the three PPI/E events, members were asked their views and recommendations on the delivery mode, and to write messages tailored to INA and NINA case studies. Data generated from PPI/E informed the content of the message file (eg, see online supplementary appendix 4). Overall, PPI/E members found more acceptable NINA case studies and seemed to generate more messages for NINA rather than INA case studies.

### Pre-test feasibility of intervention delivery mode and theoretical determinants

Four studies informed the acceptability of the delivery mode and the theoretical determinants. These included interviews, experiential focus groups and PPI, using think aloud protocols.

Face-to-face, in-depth interviews were conducted with 13 patients. Think-aloud protocol was used to assess reliability of the mechanisms of change (ie, link between the theoretical determinants and the BCTs) and usability of the delivery mode. All interviews were conducted at participants' places. Field notes and think-aloud data informed the analysis, and results refined the tailoring of the theoretical determinants.[25]

Face-to-face meetings were conducted with two PPI members recruited using emails from the list of the National Institute of Health Research (NIHR) Cambridge BRC PPI/E Panel and Communications office. Think-aloud protocol was used to assess the reliability and sequence of the items to form the measurement of the theoretical determinants (ie, theory-based questionnaire). Both interviews were conducted at the university. Field notes and think-aloud data informed the analysis, and results refined the reliability of the tailoring questionnaire and informed the decision rules of the tailoring algorithm.

Experiential focus groups aimed to obtain patients' views of, and refine, the tailored intervention content (eg, inclusion of medication name, dose and timings; 'please do not forget to take Ramipril 1 tablet 10 mg at 9 am') and delivery mode within a social interaction context.[26] Eligibility criteria and recruitment procedure were similar to patients' face-to-face interviews, reported above. Four focus groups took place in primary care practices (n=2) or community centres (n=2). During the focus groups, patients (n=12) received examples of SMS and IVR messages on their mobile phones and were asked to provide experiential feedback. Patients were asked their views and recommendations about the combination of the delivery modes and the intervention content using a think-aloud protocol.[27]

One experiential focus group with six PPI members aimed to obtain views and recommendations on the intervention content and delivery mode. PPI were recruited using emails from the NIHR Cambridge BRC PPI/E office. PPI members were eligible to take part if they were prescribed medications for a long-term health condition and were >18 years.

The data generated from the experiential focus groups informed the content of the message file and the process to map the intervention messages onto the theoretical determinants.

### Develop implementation procedures within primary care

The intervention implementation procedures were informed by systematic review evidence, PPI/E and stakeholders' consultations.

Information about the recruitment setting and patients' characteristics at baseline, as well as uptake and retention rates, was extracted from the two systematic reviews of trials on medication adherence delivered by SMS and/or IVR messages.[11 12] Due to missing data about the characteristics of the targeted population, the recruitment setting, recruitment methods and material, we synthesised the findings narratively. Results informed the decisions regarding the recruitment setting and methods, as well as the targeted behaviour and population of the intervention.

Twelve PPI members provided feedback on the recruitment material (eg, patients' information sheets, posters, leaflets, flyers and invitation letters) using emails or face-to-face meetings. Their comments informed the content of the recruitment material and the description of the recruitment, informed consent and intervention procedures in lay language.

Stakeholders' consultations were conducted with healthcare providers, commissioners and patients to inform the study design and the implementation procedures within the primary care setting (eg, recruitment, informed consent, data collection). Practitioners with experience of advising patients for medications; commissioners for medicine optimisation, prescribing or cardiovascular conditions; and patients with either HBP or both HPB and T2DM were included. Commissioners

were recruited informally by email through personal networks. Patients and healthcare providers (ie, practice nurses, healthcare assistants) were recruited through the primary care practices. Those who consented to take part were invited to the consultations. Consultation modes were flexible to maximise participation (n=3 face-to-face, n=4 email). Stakeholders (n=7) were given a description of the proposed intervention and asked their views and recommendations. Healthcare providers (n=4) were consulted about the time to initiate recruitment, and the recruitment methods and procedures. Commissioners (n=1) were consulted about how best to track resource use and costs and the evidence needed to inform whether or not to commission such an intervention. Patients (n=2) were consulted about the delivery mode and the theoretical framework. All stakeholders were asked for their recommendations about the implementation procedures. The data generated from stakeholders' consultations informed the study and intervention implementation procedures.

### Step 3: develop the prototype intervention

A prototype intervention was developed by multiple and iterative syntheses of the data obtained during the intervention development. The intervention is highly tailored to deliver very brief (≥1 min), theory-based messages at participant's preferred time, telephone and frequency to receive the intervention messages. The pilot intervention consisted of 29 messages, delivered daily for a duration of one prescription-based month. Two messages were delivered at the first day of the intervention, with feedback tailored to each participant's responses at the baseline questionnaire and prescription plan, and one message per day during the following 27 days of the intervention.

### Step 4: refine the intervention

Two PPI members, recruited by the NIHR Cambridge BRC PPI/E office, pre-tested the prototype intervention for one prescription-based month. PPI provided their experiential feedback about the intervention usability and mechanisms of change during four, weekly telephone-based interviews with a researcher, and completed the follow-up questionnaire. The data informed the decision rules of the tailoring algorithm and refined the frequency (ie, message schedule), combination and sequence of the BCTs (ie, integration of theory into the intervention), before piloting the prototype intervention within primary care.

### Intervention piloting

A 1-month pilot intervention was conducted to assess (a) implementation procedures; (b) uptake, retention, fidelity and engagement with the intervention; (c) participants' views, understanding and actions on the intervention content; and obtain (d) recommendations for improvement.

Patients with HBP were recruited by primary care practices within the Cambridgeshire and Peterborough CCG

and healthcare providers who consulted patients for taking medications. Patients were selected and recruited using the same recruitment procedure described in the intervention development studies but included two additional recruitment strategies: (a) face-to-face recruitment by healthcare providers and (b) leaflets and posters in the practice waiting rooms. Healthcare providers received training by a researcher on the face-to-face recruitment and signposting procedures. Patients completed measures at baseline (T1) and at the end of the intervention (T2).

Patients completed the tailoring questionnaire (T1, T2), their preferred primary and secondary telephone number to receive the intervention messages, and the time intervals of repeated calls per telephone number (T1), two items measuring adherence (T1, T2), 16 items measuring their experience and satisfaction with elements of the intervention (T2), and face-to-face interviews with a member of the research team (T2). Medication collection data for the duration of 3 months before T1, and at T2, were obtained from practice dispensary records. Telephone log files assessed uptake, retention, fidelity and engagement with the intervention. Participants' inbound calls assessed intervention engagement and impact. Face-to-face interviews (T2) assessed users' understanding and actions on the intervention and obtained recommendations for improvement. Face-to-face interviews with healthcare providers (n=3) assessed implementation procedures and obtained recommendations for improvement.

### Analysis

Data generated from the intervention development and pilot study were integrated into one mixed-methods analysis,[27] using the technique called 'following a thread'.[28 29] The theoretical underpinnings of the intervention were explored across different methods of data collections, data generated from each method informed each intervention component (ie, intervention content and implementation procedures) iteratively and questions or contradictory findings generated were followed across the other methods until saturation was achieved.

### RESULTS

A summary of the outcomes of the mixed-methods analysis is reported in table 1 and explained in more details in this section.

### Theory-based questionnaire

Participants' feedback on the questionnaire suggested that self-reported measures of non-adherence have confounded effect on measures of psychological constructs of adherence, and thus an objective measure of adherence was recommended. For example, Medication Adherence Report Scale items were perceived to measure INA/NINA constructs, rather than taking or not taking medications.

### Tailoring

Participants' input suggested amendments to the tailoring algorithm, to increase the sensitivity of the algorithm in selecting INA participants. For example, the cut-off values of the algorithm were amended to facilitate the identification of INA and NINA. Moreover, participants' input suggested that the theoretical determinants could be the same for INA and NINA but they might differ in terms of the value of tailoring, thus we adjusted the theoretical model and cut-off values, so that each of the theoretical determinant could appear to both INA and NINA participants (eg, see online supplementary appendix 5 for examples of tailoring BCTs).

### Message file

Participants suggested content that informed the intervention BCTs, so that we included more BCTs, such as information about social and environmental consequences.

> Do you, sort of, remind them of the costs and things? Is that in with medication and things, if they're missing it, or every time they lose blood pressure medication or throw it in the bin, what is the cost? Practitioner consultation

### Schedule

The 1-month intervention schedule included daily messages and participants could change the frequency (i.e. more or less) or stop them. Patients at follow-up interviews recommended an intervention of longer duration with messages to gradually decrease in frequency and include advice and support for self-monitoring medication taking.

### Targeted behaviour and participants' characteristics

Patients reported that it was easier for them to report on the pills taken per day rather than the specific type of

---

**Table 1** Outcomes and results from intervention development and piloting

| Intervention development | Intervention piloting |
|---|---|
| ► Theory-based questionnaire | ► Recruitment and retention rates |
| ► Tailoring algorithm | ► Fidelity and engagement with the intervention |
| ► Message file | ► Understanding and actions on intervention material |
| ► Message schedule | |
| ► Delivery mode | |
| ► Targeted behaviour and participants' characteristics | |
| ► Implementation procedures within the primary care | |

tablets (eg, felodipine, statins). However, they reported that this information could be useful for patients who initiate medication taking and need more education about their prescribed regimens. Patients reported that the timing of medication-taking or the routine associated with the timing (eg, lunch), rather than the type of medications to take, was an easier reminder of behaviour. That was particularly important for those patients taking medication for longer and for more than one health conditions.

Q: 'How many tablets do you take per day?' R: 'Oh, about 17'. Q: 'How many different types of medication do you take per days?' R: 'Yeah. That takes more counting and a bit more thinking … I suppose, as well, if you have so many, you just get used to the times … Morning, lunch, evening, night time' patient.

Patients who took part in the intervention development studies and the pilot study were recruited from different areas of IMD, were older adults, had either HBP or comorbidities (see online supplementary appendix 6 for participants' demographic characteristics) and self-reported being prescribed complex medication regimens (number of tablets per day: mean=6, SD=4.6).

### Implementation procedures within primary care

Healthcare providers that conducted the recruitment procedures reported challenges with recruiting non-adherent patients and recommended recruitment strategies delivered at times when patients might be more receptive to uptake such an intervention. In accordance with the systematic review evidence, it was recommended that patients would be more receptive to uptake the intervention when they initiate medication taking or when there is a change to their current prescription plan (eg, this could be either due to a health event and/or due change to the pharmaceutical product). Otherwise, patients could be reluctant to provide consent and uptake the intervention (table 2).

Healthcare providers recommended that recruitment methods could be integrated to annual reviews or medication reviews. They also recommended that non-adherent patients, who do not attend the practice for their annual or medication reviews, either because they do not appreciate the need of these or they are sceptical about the recommended treatment, could be more receptive to the intervention if invited by other methods; like notes into their prescriptions, text messages, newsletters or multimedia (table 2).

However, healthcare providers also revealed that time constrains prevented them from explaining the implementation procedures to eligible patients during primary care consultations. It also prevented them from facilitating patients' motivation to medication adherence and uptake to the digital intervention.

In line with data from the PPI consultations and patients' interviews, healthcare providers suggested that there is a lot of information in the recruitment material and suggested that more emphasis is needed to patient's consent process. It was suggested that patients should be made aware about what information will be collected by their practice records and how these will be used from the intervention. To facilitate patients' understanding and standardise the implementation procedures, including the informed consent, we developed videos aligned to written material. The content of the videos aimed to also provide normative information about taking medication and medication non-adherence, and increase motivation to initiate behaviour change. The videos were integrated into the study invitation material (eg, text message invitations to eligible patients) and brief consultations with the healthcare providers.

| Table 2 | Healthcare providers' experiential feedback |
|---|---|
| **Theme** | **Quote** |
| Views about recruiting non-adherent patients | '[patients who do not adhere to medication] they're just not coming in for their monitoring … a lot of the names on the list [of eligible participants] were people that weren't coming in anyway and that was the biggest problem.' Practice nurse |
| Views about barriers to recruit for a digital intervention | ''cause they thought it was them, they had bad blood pressure or something … there was a question about what other medication do you take as well, and they're like, "Why do we have to answer that if it's only about the blood pressure one?" … but he felt like they were invading his space, sort of thing, by putting down all the medications, you can kind of guess what they're for, and that's what he didn't really want to.' Practice nurse |
| Views and recommendations about recruitment methods | Annual reviews: 'Annual review is a bit more time … when its annual reviews because we're actually talking about their tablets.' Practice nurse recommendation about recruitment procedures<br>Practice website: 'But you can put a facility on the online forum. A lot of people access our website, so there are quite a lot of other places that people might look maybe more than a poster.' Practice nurse<br>Medication prescriptions: 'But every time you get a repeat prescription list, you can write something on it.' Practice nurse |

n=8 healthcare providers: nurses and healthcare assistants. Themes have been coded at interviews with healthcare providers at development (n=5) and piloting of the intervention (n=3). Quotes reported in table are from experiential interviews with healthcare providers (T2).

To reinforce the uptake and integration of the intervention into current NHS, commissioners, patients and healthcare providers reported that intervention needs to be of low or no cost. 'Would need to be very low cost or free to use' commissioner consultation.

### Intervention uptake and retention

Twenty patients were recruited. In total, 18 provided written informed consent and registered into the study, of whom 17 completed the 1-month intervention: one patient was excluded by the research team before registration because he/she met one of the exclusion criteria, one patient withdrew before registration to the intervention and the other participant dropped out during the intervention. All 17 patients completed the measures at baseline and at the end of the intervention. All primary care practices provided completed baseline (eg, refill prescription data three months before the start of the intervention) and follow-up data, for all participants.

### Intervention fidelity and engagement

During the 1-month intervention, an average of 29 messages were scheduled, of which 22.52 were received by patients. On average, 37 calls were made, of which 2.8 failed to be made due to technical issues and 13 calls were made but failed to go through because they were not picked up by the participants. On average, three calls were repeated and received by participants. Also, 10 out of the 17 participants had chosen the calls to be repeated if not answered, and 5 out of the 17 participants had provided a secondary number for the calls to be repeated, if there was no answered at the primary number. Overall, patients were engaged with the intervention and made inbound calls to change the schedule of the messages or report about intervention content. In summary, there was a good fidelity of the intervention BCTs, with the majority of the BCTs having high fidelity scores (table 3).

### Understanding and actions on intervention content

Participants reported that the intervention content increased their awareness about medication adherence and the risk and benefits of maintaining adherent to prescribed medications for long term, reinforced social support and habit formation, and reminded them to take medications as prescribed (table 4, online supplementary appendix 7).

Participants suggested that the intervention was acceptable and easy to use, and they provided their views about specific features of the intervention delivery mode and intervention content. Specifically, the tailored schedule of the messages, the personalisation and the variation of the content were found to be particularly appealing and were perceived to promote engagement with the intervention (table 5, online supplementary appendix 7).

Participants also recommended that some messages could be better received if delivered by either IVR or SMS, depending on the timing and the content of the messages. For example, they recommended text messages

**Table 3** Fidelity and engagement, intervention content

| Messages, including one or more BCTs | Scheduled | Received |
|---|---|---|
| Tailored to baseline questionnaire, integration of tailored BCTs | 1 | 0.6 |
| Personalised | 29 | 22.52 |
| Information about health consequences | 5 | 4.41 |
| Information about emotional consequences | 1 | 0.65 |
| Action planning, implementation intentions | 4 | 4 |
| Report whether or not the behaviour was performed | 2 | 1.74 |
| Social reward | 2 | 1.4 |
| Habit formation | 24 | 19.58 |
| Social support (unspecified) | 3 | 2.29 |

n=17 patients. Data reported as average number. The average number of the single BCTs reported in this table excludes the number of BCTs included in the 'tailored to baseline questionnaire, integration of tailored BCTs'.
BCTs, behaviour change techniques.

at the time of medication taking with simple reminders of the behaviour to reinforce habit formation; whereas they recommended voice messages at a time of their preference with information about how to set up a routine with medication taking. They also suggested messages that target patients' INA beliefs and coping plans, to be delivered less frequently than reminders of the behaviour, as they were more personal and therefore easier to be perceived as invasive.

> So maybe a text message … 'cause you're waiting for the phone and you've got to do something, which, you know, I think is a good thing, in that you've got to do something and you have to respond. You can't just put your phone in your pocket and walk away, but at night time, I think it would probably be a bit intrusive. Patient 01062T, follow-up interview

Participants reported being satisfied with the overall experience with the intervention, the availability of the intervention 24/7 and their ability to call in and leave a message (see online supplementary appendix 8). They would also recommend the intervention to other people who take prescribed medications (see online supplementary appendix 7).

## DISCUSSION
### Principal findings

A highly tailored intervention has been developed to support adherence to medication prescribed for high blood pressure and comorbidities. The intervention has been systematically developed based on guidance from MRC framework, PPI/E and is based on rigorous

**Table 4** Participants' experiential feedback about the intervention content

| Elements of the intervention | Themes and quotes |
|---|---|
| Message content and tailoring *Understanding and acceptability of the theoretical determinants* | **Increase awareness of the importance to take medications as prescribed** 'So, that message came over quite clear, you know, that you must—you mustn't miss them. You must take them, you know. So, I thought it was quite good.' Patient 01001X **Increase awareness of the benefits to keep taking medications as prescribed and potential risks when not taking medications** Q. Can you remember a specific message? R. 'One message said how important it was to keep taking your pills to keep you fit and healthy.' Patient 02028K R. 'One [message] talked about the benefits of taking your medication regularly.' Patient 02076E R. 'If I'm honest, no. I've had a holiday in-between. No, I don't—I think probably, the one about—there was one about keeping—you're taking it to keep well, yeah, probably that one, that's the one that sticks in my mind.' Patient 01051L **Remind to take medications** 'For me, the short message was ideal, you know? Who am I? and yes, and, "Have you taken your medication?"' Patients 01006W 'I liked the idea of just a phone call to say, "Is that {name}?" "Yeah." "Take your tablet." Straightforward, you know.' Patient 01001X 'Oh, well, just the, you know, jogging the memory … Yeah, it's, you know, the telephone rings, somebody comes out with a message, are you so and so? Don't forget to take your tablets and so, you take them.' Patient 01025G 'I will say I liked it because it reminded me, like helped me to remember to take my tablet.' Patient 01051L 'I thought it was a good system to—it just reminded me to take my tablets.' Patient 02028K **Social support** 'It was interesting to do it and to think someone's taking some notice of what I'm doing, trying to get rid of my high blood pressure.' Patient 01024X *Habit formation* 'Well, the telephone call comes through at the same time every morning and so, you know you're going to take the pills.' Patient 01025G 'Well, I mean, yes, I mean, the phone call alone, without any message at all, would remind you.' Patient 01031Q 'It helped the routine, I mean, that was just that reminder of doing things … by the end I'd developed a bit of a routine that I'm sitting there with the kind of phone ready for {time} o'clock, right, and then I would go and take my tablets.' Patient 01057V |

Data from patients' (n=17) face-to-face interviews at the end of the 1-month pilot intervention (T2).

theory and evidence. It includes a combination of BCTs tailored to patients' beliefs (eg, beliefs about medications, self-efficacy, social norms) and prescription plan. The pilot study suggested that the intervention content increased awareness about the necessity of medication adherence, the risk and benefits of maintaining adherence to prescribed medications, reinforced social support and habit formation, and reminded patients to take

**Table 5** Participants' experiential feedback about the intervention delivery mode

| Theme | Quote |
|---|---|
| Usability of the intervention, delivery mode, outbound and inbound calls | 'It was certainly easy to use. There's nothing complex about it really. I mean, if you've got any questions, you can just ring the numbers. I didn't have any questions, personally, but if you've got any, you can ring the number and it's quite easy and straightforward.' Patient 01006W 'It was always dead on time too, and if I—so, it didn't recognise it, they phoned back, sort of, about five min later and then it was alright.' Patient 01024X 'Well, it was very easy, just—yeah, very easy. I had clear numbers to call if there was a problem, which I did, on a couple of occasions. No, it was very easy … I think having the choice of time is best … Yes, yeah, no, I think that is important, the choice of time, definitely.' Patient 02028K 'It was easy and as long as it was at a set time, which it was, and there's—and if there's a second mobile phone, so if the first one misses, at least you can get hold of the person on the mobile if they're out and about.' Patient 01025G |

Data from patients' (n=17) face-to-face interviews at the end of the 1-month pilot intervention (T2).

medications as prescribed. To our knowledge, this is the first medication adherence intervention delivered using automated text and voice messages that has been developed and piloted within the UK primary care setting. The intervention has been proven to be acceptable and feasible to adults with high blood pressure or comorbidities, and it is currently being evaluated in a randomised controlled trial (RCT).

## Strengths and limitations of this study

To date there are studies describing the systematic development of mHealth interventions; however, none of them has provided a detailed description based on guidance from the MRC framework, and none has described the methods used to develop and pilot the implementation procedures within the primary care setting. Furthermore, this study has involved PPI/E in the intervention development, design and pilot, adapting a collaborative approach, which minimised the influence of the social desirability effect on the findings, and thus it provided a real-world perspective. As a consequence, the results of this intervention may be better informed and matched to patients' healthcare needs.

A strength of this study is the systematic approach to intervention development and piloting. The study used a rigorous theoretical approach to guide the development of the tailored intervention content. It also designed a flexible and scalable application to deliver messages to support adherence to different type, and complex prescribed medications. Another strength of this study is the utilisation of a mixed (eg, qualitative, quantitative) and multiperspective (patients, healthcare providers, PPI, stakeholders) methods of data collection. The data were combined complementarily into one comprehensive analysis using a rigorous analytical approach.

Although this intervention is highly tailored, and participants can provide real-time input and further tailor intervention content and delivery, a limitation of the intervention is the self-reported theoretical determinants of medication adherence. Future studies could enable the use of objectively measured theoretical determinants and objectively measured behaviour, using sensing technology. Another constrain of the text and voice messaging is its limited features to facilitate delivery of BCTs visually (eg, picture of tablets, graphs to show levels of adherence),[30] and future studies could usefully explore whether and how these features could influence intervention engagement and behaviour change.

## FUTURE RESEARCH

The 1-month intervention has been extended to 3-month intervention. It also includes another level of tailoring, where participants select between two BCTs to support maintenance of medication taking (eg, habit formation or self-monitoring). The intervention is currently being evaluated in an RCT, which includes objective behavioural (medication event monitoring system) and clinical (systolic blood pressure, HbA1c) outcome measurements to inform the results about the efficacy of the intervention (see https://doi.org/10.1186/ISRCTN10668149).

**Contributors** SS conceived the studies. AK designed the studies, developed the intervention, and led the intervention piloting, data collection, analyses and wrote this manuscript. VH assisted in data collection. SE and JB provided IT expertise in developing the digital platform. SS provided guidance and consultation to AK during all stages of the intervention development and piloting process.

**Funding** This paper presents independent research funded by the National Institute for Health Research (NIHR) under its School of Primary Care Research [grant number: RG86506] and the Research for Patient Benefit programme [grant number PB-PG-0215-36032]. The views expressed arethose of the authors and not necessarily those of the NHS, the NIHR or theDepartment of Health and Social Care.

**Disclaimer** The views expressed are those of the authors and not necessarily those of the NHS, the NIHR or the Department of Health.

**Competing interests** None declared.

**Patient consent for publication** Obtained.

**Ethics approval** The Office for Research Ethics Committees of Northern Ireland (ORECNI; REC references: 16/NI/0150) and Essex East of England (REC Reference number 17/EE/0203), and the Health Research Authority, have reviewed and approved this project.

**Provenance and peer review** Not commissioned; externally peer reviewed.

**Data sharing statement** Qualitative and quantitative data collected for the purposes of this study are available upon request to AK.

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
