## [Reviewer comments · BMJ Open]

ARTICLE DETAILS

TITLE (PROVISIONAL)	Development and piloting of a highly tailored digital intervention to support adherence to anti-hypertensive medications, as an adjunct to primary care consultations.
AUTHORS	Kassavou, Aikaterini; Houghton, Vikki; Edwards, Simon; Brimicombe, James; Sutton, Stephen

VERSION 1 – REVIEW

REVIEWER	Ayeesha Kamran Kamal Aga Khan University Karachi, Pakistan
REVIEW RETURNED	25-May-2018

GENERAL COMMENTS	Well written with attention to methodology , for contextual reasons discuss your qualitative research methodology eg theme saturation etc and identification of factors with FGDs what methods you followed. COREQ guidelines may assist.
---

REVIEWER	Ian M. Brooks Associate Director, Biomedical Informatics, MedStar Health Research Institute, 6525 Belcrest Road, Suite 714, Hyattsville, MD 20782 USA
REVIEW RETURNED	26-Jun-2018

GENERAL COMMENTS	This is an interesting and detailed manuscript, and ties in with the current knowledge in the field. The authors are adding to the body of evidence for this style of intervention/approach. In addition to the expected patient engagement interviews for developing/refining an approach, the lead author has also conducted a series of interviews with various clinical stakeholders, including practice nurses who are the front line of our patient engagement efforts; this is a strong positive. I just wish data were presented on the views of the stakeholders, highlighting points of confluence, and how they might tie in with the comments/needs/expectations of the patient interviewees. My specific notes for consideration: 1. The manuscript needs to be reviewed by a scientific copy editor for grammar, spelling, and narrative flow. There are multiple small issues that cause reading hiccups that break the narrative and force the reader to revisit paragraphs, or shuffle back-and-forth trying to tie/synch ideas up.2. Abbreviations need to be spelled out (MRC, BMC, NICE), and please consider adding a small series of notes to help non-UK readers negotiate the intricacies of the NHS/MRC etc. For example, what is the MRC framework & why is it relevant that this study follows it so closely; why (aside from the obvious) does the
---

	Commissioner expect any embedded intervention to be free/low cost; could it be rolled out at specific sites/Trusts etc.? Not necessarily to the extent of a true cost/benefit analysis, but what potential savings could be expected if the intervention was added as part of the standard of care? 3. More data please - for example, are the intervention development subjects HBP/T2D? What are their demographic characteristics and do they map to the larger at-risk population, and how do they compare to the pilot trial subjects. Same for the 20 enrolled subjects in the trial. There is a large amount of text in the Methods that could perhaps be abbreviated and more information included in the Results. 4. Table 1 should be in the Results section, not the Methods. Larger issues:  1. The tailoring component...are you referring to tailoring the message content/subject based on their interviews; or tailoring the delivery method (IVR and framework)? Or both? This must be clarified. 2. In the Discussion you mention that this is the first trial to use IVR and SMS, but the Methods and Results talk only of IVR/calls. Please clarify. 3. The data in Appendices 6 & 7 seem to contradict your reports of success. If 5 is the lowest value in the Likert scale, then a satisfaction report of 4+ would indicate unsatisfactory, no? 4. Pg 10, paragraph 3, "Due to missing data we synthesized the findings narratively" – please explain; this entire paragraph is hard to parse. 5. For both the test intervention and the pilot trial – what fraction of patients had INA vs NINA issues? Was message content and frequency tailored to address these? Did patients have 100% control over the content of the messages? This is important to make sure that selection biases don't abrogate potential successful intervention into non-adherence issues.
--	--

VERSION 1 – AUTHOR RESPONSE

Reviewer: 1

Reviewer Name: Ayeesha Kamran Kamal

Institution and Country: Aga Khan University, Karachi, Pakistan

Please state any competing interests or state 'None declared': None declared

Please leave your comments for the authors below

Comment 1. Well written with attention to methodology , for contextual reasons discuss your qualitative research methodology eg theme saturation etc and identification of factors with FGDs what methods you followed. COREQ guidelines may assist.

Response: Many thanks for your comment. I assume that 'FGDs' this refers to 'focus groups' ? if yes, then the method followed to collect data is reported at page 9, paragraph 3, section 'Pre-test feasibility of intervention delivery mode and theoretical determinants' (e.g., 'think aloud protocol'). Discussion about data saturation is reported at page 12, analysis. However this refers to saturation of data collected to respond to the research questions, not the qualitative studies, for the reasons reported at the response to you recommendation below.

Many thanks for your recommendation to include the COREQ guidelines for interviews and focus groups. Some of the criteria included in this guideline have facilitated the reporting of the qualitative results in this paper. However, detailed reporting of the qualitative methods and results could not be included in this paper because that was not the aim of this research. This research treated qualitative

studies as methods to collect data and inform intervention development aims and objectives (e.g., pre-test acceptability of the intervention delivery mode and intervention content), rather than as primary studies to provide insight into how participants make sense of their medication taking experience. Instead, this research used the TREND statement checklist to facilitate reporting of studies in this paper.

Reviewer: 2

Reviewer Name: Ian M. Brooks

Institution and Country: Associate Director, Biomedical Informatics, MedStar Health Research Institute, 6525 Belcrest Road, Suite 714, Hyattsville, MD 20782 USA

Please state any competing interests or state 'None declared': None declared

Please leave your comments for the authors below

This is an interesting and detailed manuscript, and ties in with the current knowledge in the field. The authors are adding to the body of evidence for this style of intervention/approach. In addition to the expected patient engagement interviews for developing/refining an approach, the lead author has also conducted a series of interviews with various clinical stakeholders, including practice nurses who are the front line of our patient engagement efforts; this is a strong positive. I just wish data were presented on the views of the stakeholders, highlighting points of confluence, and how they might tie in with the comments/needs/expectations of the patient interviewees.

My specific notes for consideration:

Comment 1. The manuscript needs to be reviewed by a scientific copy editor for grammar, spelling, and narrative flow. There are multiple small issues that cause reading hiccups that break the narrative and force the reader to revisit paragraphs, or shuffle back-and-forth trying to tie/synch ideas up.

Response 1. Grammar and spelling has been checked by an English speaking editor. To facilitate narrative flow and sync of ideas, the reviewer could possibly look at Appendix 1 and Appendix 2, which provide an overview of the main ideas and sync these up. The manuscript has also been reviewed by a scientific copy editor.

Comment 2. a) Abbreviations need to be spelled out (MRC, BMC, NICE), and please consider adding a small series of notes to help non-UK readers negotiate the intricacies of the NHS/MRC etc. b) For example, what is the MRC framework & why is it relevant that this study follows it so closely; c) why (aside from the obvious) does the Commissioner expect any embedded intervention to be free/low cost; d) could it be rolled out at specific sites/Trusts etc.? e) Not necessarily to the extent of a true cost/benefit analysis, but what potential savings could be expected if the intervention was added as part of the standard of care?

Response 2.

a) Abbreviations were spelled out.

b) MRC framework has been used for developing complex intervention and is recommended by the Department of Health as a framework to develop and evaluate complex interventions.

c) When developing the commissioners' consultations material, we have not included a question about the reasons why commissioner expect these interventions to be free or low cost, because that was not part of the research plan. Although very interesting idea for future research, more information on this subnote cannot be provided by this study.

d) The intervention could be rolled out at specific sites, but evidence from a larger trial are needed to provide rigorous data and a response to this subnote.

e) Again, the purpose of this paper was to present evidence on the process to develop and pilot the intervention. Information about the potential savings that the intervention could provide, if it was part of the standard care, can only be provided from a randomised controlled trial design. To authors knowledge, there is no previous study in the UK to provide evidence about the potential savings of this intervention. However, evidence from a cost-effectiveness trial conducted in Australia has

reported that such interventions could lead to 563 fewer myocardial infarctions, 361 fewer strokes and 1143 additional QALYs, and cost savings of 10.56 million for the healthcare of the patients' lifetimes. Thus, such interventions have the potential to be health improving and cost saving. A sentence with this information has been included in the manuscript page 4, last paragraph.

Comment 3. More data please - for example, are the intervention development subjects HBP/T2D? What are their demographic characteristics and do they map to the larger at-risk population, and how do they compare to the pilot trial subjects. Same for the 20 enrolled subjects in the trial. There is a large amount of text in the Methods that could perhaps be abbreviated and more information included in the Results.

Response 3. The demographic characteristic of the patients that took part in the intervention development studies and the pilot intervention are presented at results, page 14, targeted behaviour and participants' characteristics, at appendix 6, and at page 20, the first paragraph of the discussion.

Larger issues:

Comment 1. The tailoring component...are you referring to tailoring the message content/subject based on their interviews; or tailoring the delivery method (IVR and framework)? Or both? This must be clarified.

Response 1. The tailoring component refers to tailoring the intervention content to the theoretical determinants (see page 6, second paragraph 'tailor the theoretical determinants').

Comment 2. In the Discussion you mention that this is the first trial to use IVR and SMS, but the Methods and Results talk only of IVR/calls. Please clarify.

Response 2. At discussion I mention that 'this is the first medication adherence intervention delivered using automated text and voice messages that has been developed and piloted within the UK primary care setting', not that this is the first trial to use IVR and SMS.

At methods I talk about the development of both IVR and SMS application (see page 7, paragraph 1, 'A custom written database application contained the functionality to create output for both delivery modes, and data were transferred at regular intervals to both platforms (IVR and SMS) using secure data hosting protocols'); not of IVR/calls.

At results more quotes about the IVR calls were included, because participants tended to talk more about this element for the intervention.

Comment 3. The data in Appendices 6 & 7 seem to contradict your reports of success. If 5 is the lowest value in the Likert scale, then a satisfaction report of 4+ would indicate unsatisfactory, no?

Response 3. it seems that there was a typo in the footnote of these tables, I am sorry. Number 5 refers to 'strongly agree' or 'very satisfied', whereas the 1 refers to 'strongly disagree' or 'very dissatisfied'. Thus the results do not contradict reports. I have amended the notes at appendixes.

Comment 4. Pg 10, paragraph 3, "Due to missing data we synthesized the findings narratively" – please explain; this entire paragraph is hard to parse.

Response 4. The paragraph has been rephrased to explain that data regarding the setting, methods, and participants' characteristics in primary studies included in reviews were missing. Thus quantitative analysis to identify the most effective setting and methods of recruitment, as well as participants' characteristics at baseline and follow up, could not be performed.

Comment 5. a) For both the test intervention and the pilot trial – what fraction of patients had INA vs NINA issues? b) Was message content and frequency tailored to address these? c) Did patients have 100% control over the content of the messages? This is important to make sure that selection biases don't abrogate potential successful intervention into non-adherence issues.

a) the determinants for the intervention were only measured for the pilot study, not the intervention development studies. The majority of the participants had NINA.

b) Content. message content were tailored to address INA, NINA (see page 6, paragraph 2, line 26). Frequency. frequency of the intervention content was tailored to INA, NINA; however for this pilot study the frequency of the delivery mode was not tailored to INA, NINA (see page 6, paragraph 3, line 42).

c) Patients had 100% control over the content of the messages, they could change the content of the messages at any time (see page 6, paragraph 2, line 31-35).

VERSION 2 – REVIEW

REVIEWER	Ayeesha Kamran Kamal Aga Khan University Karachi, Pakistan
REVIEW RETURNED	31-Jul-2018

GENERAL COMMENTS	This is a very well done study, the authors can also go through the new CONSORT e Health checklist prior to final publication to further refine their intervention.
---

REVIEWER	Ian M. Brooks, Ph.D. Associate Director, Biomedical Informatics, MedStar Health Research Institute, 6525 Belcrest Road, Suite 714, Hyattsville, MD 20782, USA
REVIEW RETURNED	27-Aug-2018

GENERAL COMMENTS	No additional comments. Thank you to the author for detailed responses.
---